# SARS-CoV-2 infection induces sustained humoral immune responses in convalescent patients following symptomatic COVID-19

Jun Wu [1,2,12], Boyun Liang[1,2,12], Cunrong Chen[3,12], Hua Wang[1,2,12], Yaohui Fang[4], Shu Shen [4], Xiaoli Yang[1,2], Baoju Wang[1,2], Liangkai Chen [5,6], Qi Chen[7], Yang Wu[7], Jia Liu[1,2], Xuecheng Yang[1,2], Wei Li[1,2], Bin Zhu[1,2], Wenqing Zhou[1,2], Huan Wang[1,2], Sumeng Li[1,2], Sihong Lu[1,2], Di Liu[8], Huadong Li[9], Adalbert Krawczyk [10], Mengji Lu[2,11], Dongliang Yang[1,2], Fei Deng [4✉], Ulf Dittmer[2,11✉], Mirko Trilling [2,11✉] & Xin Zheng[1,2✉]

Long-term antibody responses and neutralizing activities in response to SARS-CoV-2 infection are not yet clear. Here we quantify immunoglobulin M (IgM) and G (IgG) antibodies recognizing the SARS-CoV-2 receptor-binding domain (RBD) of the spike (S) or the nucleocapsid (N) protein, and neutralizing antibodies during a period of 6 months from COVID-19 disease onset in 349 symptomatic COVID-19 patients who were among the first be infected world-wide. The positivity rate and magnitude of IgM-S and IgG-N responses increase rapidly. High levels of IgM-S/N and IgG-S/N at 2-3 weeks after disease onset are associated with virus control and IgG-S titers correlate closely with the capacity to neutralize SARS-CoV-2. Although specific IgM-S/N become undetectable 12 weeks after disease onset in most patients, IgG-S/N titers have an intermediate contraction phase, but stabilize at relatively high levels over the 6 month observation period. At late time points, the positivity rates for binding and neutralizing SARS-CoV-2-specific antibodies are still >70%. These data indicate sustained humoral immunity in recovered patients who had symptomatic COVID-19, suggesting prolonged immunity.

[1] Department of Infectious Diseases, Union Hospital, Tongji Medical College, Huazhong University of Science and Technology, Wuhan, China. [2] Joint International Laboratory of Infection and Immunity, Huazhong University of Science and Technology, Wuhan, China. [3] Department of ICU, Fujian Medical University Union Hospital, Fuzhou, China. [4] State Key Laboratory of Virology, Wuhan Institute of Virology, Chinese Academy of Sciences, Wuhan, China. [5] Department of Nutrition and Food Hygiene, Hubei Key Laboratory of Food Nutrition and Safety, School of Public Health, Tongji Medical College, Huazhong University of Science and Technology, Wuhan, China. [6] Ministry of Education Key Lab of Environment and Health, School of Public Health, Tongji Medical College, Huazhong University of Science and Technology, Wuhan, China. [7] Hubei Provincial Center for Disease Control and Prevention, Wuhan, China. [8] Pritzker School of Medicine, University of Chicago, Chicago, USA. [9] Jin Yin-tan Hospital, Wuhan, China. [10] Department of Infectious Diseases, University Hospital of Essen, University of Duisburg-Essen, Essen, Germany. [11] Institute for Virology, University Hospital of Essen, University of Duisburg-Essen, Essen, Germany. [12]These authors contributed equally: Jun Wu, Boyun Liang, Cunrong Chen, Hua Wang. ✉email: df@wh.iov.cn; ulf.dittmer@uni-due.de; mirko.trilling@uni-due.de; xin11@hotmail.com

As of January 18, 2021, the global number of confirmed cases of coronavirus disease 2019 (COVID-19) has reached 93.8 million, with more than 2,026,093 known fatalities[1]. In December 2019, the sarbecovirus severe acute respiratory syndrome coronavirus 2 (SARS-CoV-2) was identified as the causative pathogen of COVID-19. The virus has spread around the world at a rapid pace. The COVID-19 pandemic represents the greatest medical and socioeconomic challenge of our time. There is no sufficiently effective antiviral drug to treat COVID-19 cases. It is crucial for decision-making and vaccine development to understand how long immunity against SARS-CoV-2 persists in infected individuals and whether antibodies produced in response to a natural infection provide protective immunity, which may prevent reinfection with SARS-CoV-2.

To our knowledge, the observation period for most studies on SARS-CoV-2-specific antibodies is within 12 weeks[2] and it remains unclear how antibody titers may change over subsequent periods. Due to the use of different detection methods (e.g., enzyme-linked immunosorbent assay versus capture chemiluminescence immunoassays (CLIA)), the analysis of different subtypes of antibodies (immunoglobulin G (IgG), IgM, or IgA), and the focus on different antigens and epitopes (N, S, or the receptor-binding domain [RBD] of S), a coherent description of the humoral immune response after natural SARS-CoV-2 infections is not available. As has been consistently shown in short-term studies, a seroconversion of IgG and IgM occurs about 2–3 weeks after disease onset[3] and IgM levels drop significantly earlier than IgG titers[4]. However, it is unclear which antibody type (IgG or IgM) performs best in the epidemiologic identification of convalescent patients. Some authors favored IgG[3,4], while other proposed a higher positivity rate for IgM[5]. In addition, the reported peak of IgM responses was assigned to different time points ranging from 2 to 5 weeks[2,3,5].

So far, studies that analyzed only a few patients or that had an observation period of only a few weeks suggested that antibody levels may decrease rapidly in infected individuals[6]. This has been greatly discussed worldwide because it may be a very important aspect for immunity following the natural infection and vaccine development. However, long-term studies are needed because immune responses always decline after acute infections, which does not predict the duration of a protective response.

SARS-CoV-2 has a single-stranded positive-sense RNA genome which encodes structural and nonstructural proteins, including the spike (S) and the nucleocapsid (N) protein[7]. A part of the transmembrane S protein is present on the virion surface and binds to the entry receptor ACE2 mediating entry into target cells[8], while the highly abundant N protein binds to the viral RNA inside viral particles. Previous research on SARS and Middle East respiratory syndrome has shown that IgG responses recognizing S and N have different characteristics in terms of response time, duration, and titers[9,10]. In certain diseases such as dengue virus infections, binding but nonneutralizing antibodies have even been associated with worse clinical outcomes through antibody-dependent enhancement (ADE), suggesting that under certain circumstances antibodies may at least correlate with harmful effects in some patients. ADE in the context of SARS-CoV-2 has been discussed recently[11]. Higher antibodies have also been associated with older age[2] in COVID-19 patients. However, studies using pseudovirus-particle-based systems[12,13] suggest that plasma derived from convalescent patients have potent neutralizing activity that was related to IgG molecules recognizing the RBD of the S protein, suggesting that IgG-RBD-S antibodies have a high likelihood to fulfill neutralizing functions (nAbs). Some small cohort studies suggest that severe COVID-19 patients benefit from convalescent plasma (CP) therapy[14]. Very recently, highly potent SARS-CoV-2 neutralizing antibodies have been isolated and characterized from COVID-19 patients[15,16]. Thus, virus-specific antibodies seem to be very important for immunity against SARS-CoV-2 infection and/or COVID-19 development. However, it remains to be clarified how the kinetics of binding and neutralization antibodies change during and after the course of COVID-19.

In this work, we characterize the kinetics and magnitude of the initial antibody response against SARS-CoV-2 in a large cohort of symptomatic COVID-19 patients from Wuhan. Most importantly, many of these patients, who were among the first to be infected with SARS-CoV-2 worldwide, were followed up for several months to measure sustainability of the antibody response against SARS-CoV-2. We evaluate the IgM and IgG responses against the RBD of the S protein and the N protein longitudinally after the onset of symptomatic COVID-19. Presence of these antibodies and neutralizing activities of plasma were studied over 26 weeks. The results of this study provide an experimental basis for evaluating the onset and duration of humoral immunity in COVID-19 patients in order to support clinical drug and vaccine development and decision-making in terms of social-economic mitigation strategies.

## Results

**Kinetics and magnitude of the antibody response to SARS-CoV-2.** In order to investigate antibody responses toward SARS-CoV-2 over time, a total of 585 samples—obtained from 349 symptomatic COVID-19 patients—collected up to 26 weeks after disease onset were analyzed for IgM and IgG recognizing the RBD of the S protein (denoted IgM-S and IgG-S, respectively) as well as IgM and IgG binding the N protein (IgM-N and IgG-N, respectively). As test system, a capture CLIA was used.

During the initial outbreak in Wuhan, nucleic-acid-based detection methods were always complemented with antibody detection assays for the diagnosis of suspected COVID-19 diseases. All analyzed patients in this study were symptomatic for COVID-19. During the first week after symptom onset, the four antibodies were tested positive with different frequencies: IgM-S (67%) > IgG-N (33%) > IgM-N (22%) > IgG-S (11%) (Fig. 1a). The positive rate for IgM-S reached a peak of 95% at week 5 and then rapidly decreased to 0% at week 13 fluctuating below 35% thereafter. IgM-N could be detected in 72% of the patients at week 3. Afterward, this number rapidly declined and IgM-N became undetectable at weeks 10 and 12, followed by negligible fluctuations at very low positive rates. IgG-S was already positive in 98% of the patients at week 3 and remained at a relative high percentage until the end of the observation period at week 26. The positive rate of IgG-N rose rapidly to 88% of the patients at week 2 and stayed at very high levels thereafter.

We further analyzed whether a combined antibody testing may support clinical diagnostics (Supplementary Fig. 1). A combination of IgM-S and IgM-N test did not increase the sensitivity compared to IgM-S alone. Combined IgG-S and IgG-N increased the positive rate compared to IgG-S or IgG-N alone at some time points, suggesting a diagnostic benefit. In agreement with previous studies[3], the combination of IgM-S, IgM-N, IgG-S, and IgG-N resulted in positive rates approaching 100% after week 4, indicating that virtually all COVID-19 patients raise detectable humoral immune responses against SARS-CoV-2.

We also determined the dynamics of specific antibody titers during 26 weeks after symptom onset in COVID-19 patients (Fig. 1b, c). Interestingly, IgM-S and IgG-S peaked 1 week later than IgM-N and IgG-N (Fig. 1b). The titer of IgM-S reached its peak at week 4, and then slowly decreased until the average value fell below the cutoff value at week 12. After reaching the peak at week 3, the titers of IgM-N dropped rapidly below the cutoff

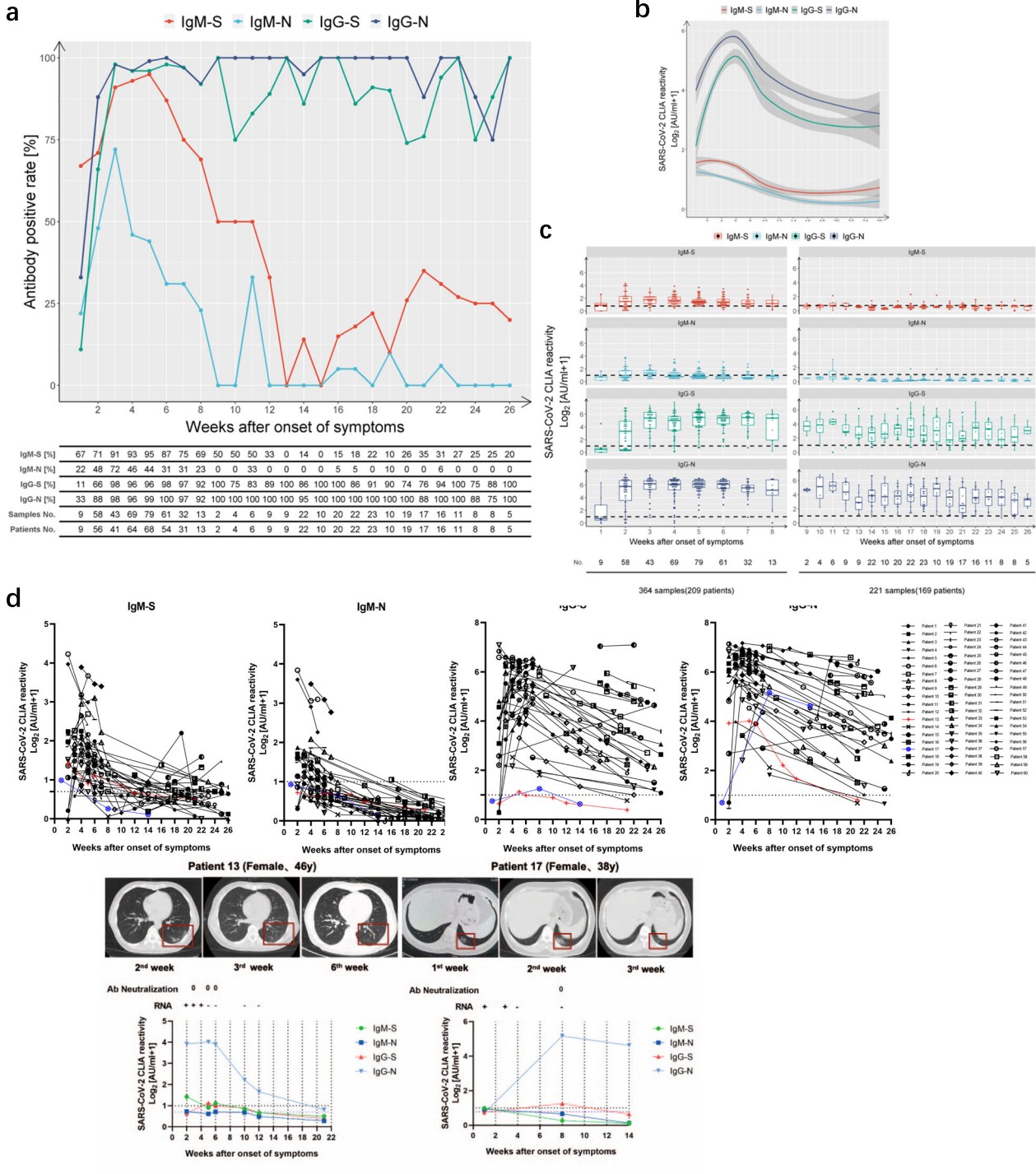

**Fig. 1 Longitudinal analyses of IgM and IgG against SARS-CoV-2 S/N in COVID-19 patients.** IgM and IgG against the RBD of the spike protein ("S") and the nucleoprotein ("N") of SARS-CoV-2 of 585 samples obtained from 349 patients were detected by capture chemiluminescence immunoassays (CLIA). **a** Positive rate of individual antibodies tested at the indicated dates following onset of symptoms. **b**, **c** The plasma antibody levels of IgM-S (red), IgM-N (light blue), IgG-S (green), and IgG-N (dark blue) in patients with different disease courses are presented. The line shows the mean value expected from a Lowess regression model, with shaded area representing 95% confidence interval in **b**. The boxes in **c** show medians (middle line), 75% quartiles (upper bound) and 25% quartiles (lower bound), and the whiskers show 1.5× the IQR above and below the box. The table below the figure represents the number of samples at each time point. **d** Sequential sampling and analyses of antibody titers in 60 COVID-19 cases. Characteristics of two patients with low IgG antibody levels. Patient 13: a 46-year-old female with fever, cough, dizziness, and fatigue for 6 days; patient 17: a 38-year-old female with fever and chest tightness for 4 days. The cutoff value for IgM-S detection was 0.7 AU/ml. The cutoff value for IgM-N, IgG-S, and IgG-N was 1 (shown on the left Y axis).

value after around 9 weeks. The titers of IgG-N and IgG-S reached their peaks at weeks 4 and 5, respectively. After a contraction phase, in which titers constantly decreased during weeks 6–14, IgG-N and IgG-S titers stabilized and were maintained at high levels until the end of the observation period 26 weeks post symptom onset. Thus, SARS-CoV-2-specific IgG responses were very similar to antibody responses against many other viruses with a peak activity a few weeks after infection,

which was followed by a contraction phase over several weeks, but finally resulting in a stabilized antibody response that could be detected for at least 6 months.

To corroborate our findings, antibody titers of 60 prototypical patients with repetitive sampling were analyzed. Except for two unusual patients (patient 13, patient 17), who did not develop measurable IgG-S response, IgM titers generally declined rapidly, while IgG titers were far more stable. Even after 26 weeks, all but three patients (patient 13, patient 14, patient 17) presented with detectable IgG antibodies (Fig. 1d, upper panel). Intriguingly, the two unusual patients who did not develop IgG-S responses were young women diagnosed with symptomatic COVID-19 accompanied with lung lesions (Fig. 1d, lower panel). Both women exhibited moderate IgG responses recognizing the N protein without developing relevant IgG-S titers. The neutralization activity of their plasma was tested at different occasions with consistently negative results, highlighting the importance of S-specific IgG for virus neutralization.

**Correlation of antibody levels with virus control and disease severity**. In order to clarify the interplay between antibodies and virus control, disease severity, gender, as well as age in COVID-19 inpatients, we compared the antibody titers amongst different patient groups. The clinical and laboratory characteristics of COVID-19 patients at the time of admission are depicted in Supplementary Table 1. Taken together, 149 (71.3%) nonsevere cases and 60 (28.7%) severe cases from isolation wards with complete medical records were enrolled. No significant differences concerning gender and age were observed between these two groups. Consistent with previous reports[17], severely ill patients showed significantly decreased counts and frequencies of lymphocytes ($p < 0.01$, two-tailed Mann–Whitney $U$ test) and decreased PLT counts ($p < 0.05$, two-tailed Mann–Whitney $U$ test) compared to patients with nonsevere disease courses, while the counts and frequency of neutrophils were increased ($p < 0.01$, two-tailed Mann–Whitney $U$ test). As expected, patients from the group with severe diseases presented with significantly increased total bilirubin, alanine aminotransferase, aspartate transaminase, lactate dehydrogenase, creatine kinase, Creatinine, D-dimers, prothrombin time, and fibrinogen than the nonsevere group ($p < 0.05$, two-tailed Mann–Whitney $U$ test).

In order to investigate the correlation between antibody responses and virus control, patients were stratified according to the presence or absence of SARS-CoV-2 RNA at the time point of antibody determination. At early time points, antibody levels were significantly higher in the group in which SARS-CoV-2 RNA was no longer detected compared to the group with prolonged SARS-CoV-2 RNA positivity. This finding strongly suggests that the presence of IgM and IgG recognizing the S and N protein of SARS-CoV-2 constitutes a clinically relevant correlate of protection in humans and contributes to virus control during the early phase of infection (Fig. 2a and Supplementary Fig. 2a).

Given the debate concerning the duration of antibody responses in asymptomatic patients[6], we wondered if nonsevere and severe COVID-19 cases might differ concerning their humoral immune responses. There were significant higher IgG-S/N responses in patients with nonsevere symptoms at week 2, again pointing toward a protective role of IgG (Fig. 2b and Supplementary Fig. 2b). The IgG-N levels of patients with severe symptoms were temporarily higher than in those with nonsevere disease at week 4 which may be a consequence of higher virus replication and antigen loads raising stronger immune responses. Accordingly, severe patients exhibited high IgG-S and IgG-N titers (Fig. 2c).

In general, life-threatening COVID-19 cases are more frequent in males and in the elderly[18]. Therefore, the relationship between gender and age with antibody levels was also investigated. Males tended to have significantly more SARS-CoV-2-specifc IgM (Supplementary Fig. 2c, d), whereas IgG responses did not show a consistent sexual disparity. At later time points, the levels of the four antibodies were significantly higher in elderly patients (≥65 years old) than those in patients younger than 65 years (Supplementary Fig. 2e, f), which might reflect higher viral loads in elderly patients.

**IgG-RBD-S titers correlate with neutralization activity**. NAbs exhibit strong therapeutic and prophylactic efficacies in SARS-CoV-2-infected hACE2-transgenic mice[19] and a recent vaccination study conducted in nonhuman primates identified NAbs as correlate of protection[20]. In order to study the duration of the neutralization capacity of antibodies, virus neutralization tests were conducted using 186 samples from 137 patients. As early as 2 weeks post symptom onset, half of the patients demonstrated neutralization activity with at 50% virus neutralization at a minimum plasma dilution of 1:20 (Fig. 3a). By week 4, the proportion of patients with neutralization activity increased to over 90%, and then remained very high until the end of the observation period after 26 weeks (Fig. 3a). Neutralizing activity at a serum dilution of 1:160 has been used as a cutoff in a clinical proof-of-concept study showing the efficacy of CP therapy[21]. A considerably high frequency of individuals in our study exhibited such strong neutralizing capacities (≥1:160). The finding that elite neutralizers (≥1:320) were not evident before week 7 suggests that it takes some time to raise very potent antibody responses.

To further determine which antibody classes and specificities may exert the neutralizing effect, correlations between the titers of the four antibodies and the neutralizing activity were analyzed. The IgG-RBD-S titer demonstrated by far the highest positive correlation with neutralizing activity (Spearman $r = 0.6932$, $p < 0.0001$), compared to IgM-S (Spearman $r = 0.2220$, $p < 0.05$) and IgG-N (Spearman $r = 0.3621$, $p = 0.0001$) (Fig. 3b). High levels of neutralizing activity (1:160 or 1:320) were only found in conjunction with high IgG-S, while plasma with high IgG-N titers or unilateral IgM responses did not correlate with high neutralizing activity (Fig. 3c). These findings are consistent with the notion that IgG-S confers neutralizing capacities.

Virus neutralization tests must be performed in BSL3 laboratories which are not broadly available. Therefore, we analyzed the receiver operating characteristic curve and the area under the curve for the IgG-S titers that are associated with virus neutralization. Titers over 4.99 AU/ml were found to constitute a threshold value to predict neutralizing effects, which may help to screen CP for immunotherapy if high level biosafety laboratories are not available (Supplementary Fig. 3). This very strict cutoff value of IgG-S titers was applied to calculate the positive rate of neutralizing activity in samples that could so far not been tested in the neutralization assay. The majority of patients were above this threshold at the latest time point of week 26, indicating the presence of IgG antibodies recognizing the RBD predictive for neutralizing activity. Please note that our very strict criteria for sensitivity and specificity underestimate the true frequency of individuals with neutralizing antibodies as can be seen when the cutoff is applied to the neutralization data set in Fig. 3b. Thus, the vast majority of COVID-19 patients raised IgG-RBD-S-binding antibodies with neutralizing capacity, which were maintained over the observational period of 6 months (Fig. 3a, d).

**Discussion**

There are tremendous global efforts by companies and academia to design, evaluate, and manufacture prophylactic vaccines against SARS-CoV-2 and the associated COVID-19. Although

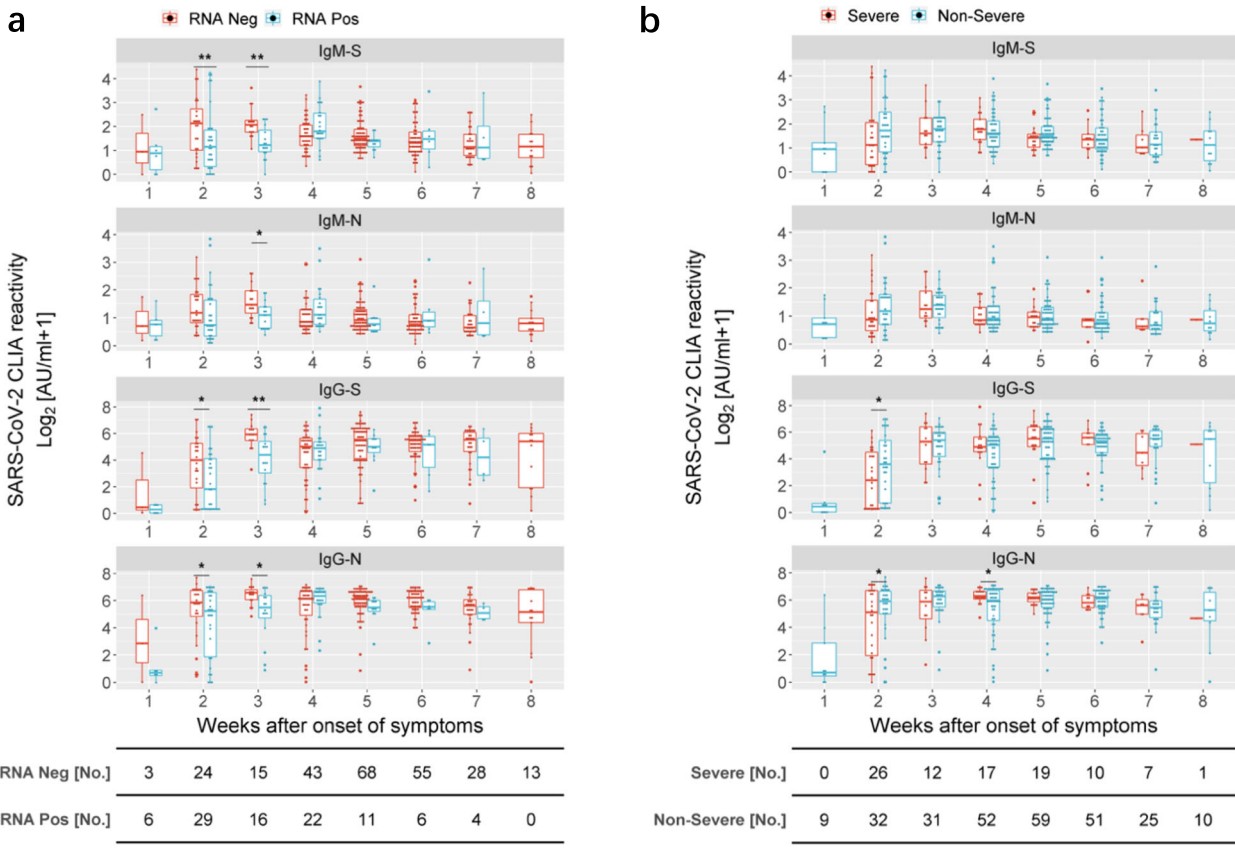

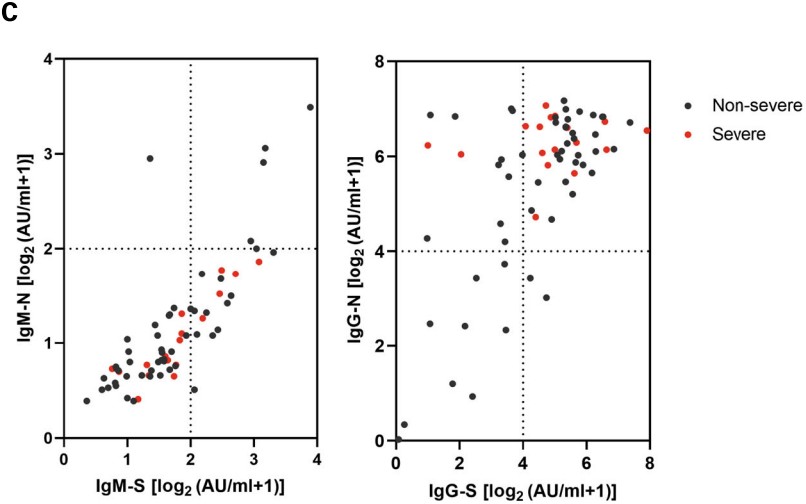

**Fig. 2 Correlation of antibody titers with virus control and severity of illness in COVID-19 patients. a** S- and N-specific CLIA-reactive IgM/IgG were compared in COVID-19 patients who were virus RNA-negative (red) versus those who were virus RNA-positive (blue) at the time point of sampling at different periods after the disease onset. A total of 343 results were acquired for this analysis. Each antibody detection value is either classified into the RNA-negative group or the RNA-positive group according to the simultaneous RNA detection result. Adjusted $p$ values are as follows: second week after onset: IgM-S ($p = 0.008$); IgG-S ($p = 0.012$); IgG-N ($p = 0.042$); third week after onset: IgM-S ($p = 0.007$); IgM-N ($p = 0.023$); IgG-S ($p = 0.005$); IgG-N ($p = 0.014$). **b** Comparison of S- and N-specific CLIA-reactive IgM/IgG titers between severe ($n = 60$, red) and nonsevere ($n = 149$, blue) patients at different periods after the disease onset. Adjusted $p$ values are as follows: second week after onset: IgG-S ($p = 0.028$); IgG-N ($p = 0.028$); third week after onset: IgG-N ($p = 0.019$). The boxes in **a** and **b** show medians (middle line), 75% quartiles (upper bound) and 25% quartiles (lower bound), and the whiskers show 1.5× the IQR above and below the box. Repeated measures (mixed model) ANOVA was used for statistical analysis. *$p < 0.05$; **$p < 0.01$; ***$p < 0.001$, two-sided. The table below the figure represents the number of samples at each time point. **c** Comparison of 64 severe and nonsevere patients (69 samples) at different S- and N-specific CLIA-reactive IgM/IgG levels at the fourth week after symptoms onset.

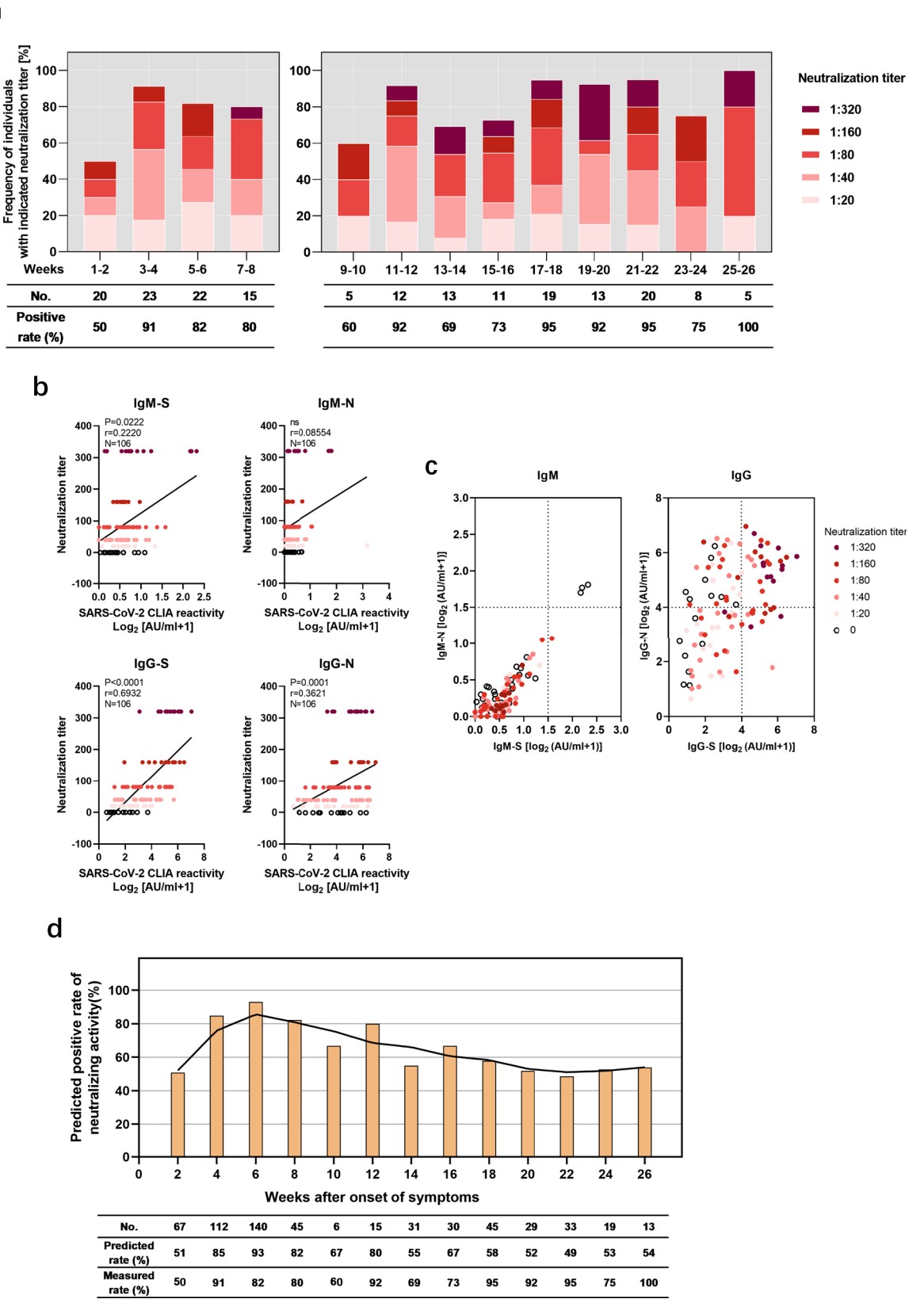

**Fig. 3 N-specific and S-specific IgG responses have different predictive values for neutralization.** A total of 186 samples from 137 symptomatic COVID-19 patients were assessed concerning SARS-CoV-2 neutralization titers and grouped according to the weeks after symptom onset. **a** Proportions of plasma neutralization activity were stratified in 2-week intervals. **b** Correlation analysis of neutralization titer with S- and N-specific CLIA-reactive IgM/IgG in COVID-19 patients. A nonparametric Spearman's correlation test was used for the statistical analyses. In the graphs, *p*, *r*, and *n* indicate the *p* value, correlation coefficient, and sample size, respectively. **c** Distribution of neutralizing activity at different S- and N-specific CLIA-reactive IgM/IgG. **d** based on the predicted cutoff value and IgG-S titer, the neutralizing activity of all confirmed patients at different time points was calculated.

such a vaccine is obviously highly desirable and first efficacy studies in animals and safety studies in humans appear very promising, there is no guarantee that a vaccine will be broadly available soon and that their protection is long lasting. Another issue is the question, if a natural infection raises a sustained protective immunity, enabling the establishment of collective herd immunity. In both cases, the duration of antibody responses is of critical relevance. At present, the sustainability of protective immunity of convalescent COVID-19 patients is one of the most urgent issues. If existent, convalescent individuals could benefit from their immunity to serve at system relevant positions and long-lasting immunity would also increase the public confidence in vaccines. To the best of our knowledge, with 6 months, the observational period of our study on the dynamics of antibody responses is the longest so far. We found that SARS-CoV-2-specific IgM recognizing S and N was only transient and disappeared around week 12. Thus, IgM responses will most likely not contribute to sustained immunity against SARS-CoV-2. We even did not find any clear correlation between IgM responses and the ability of plasma to neutralize virus in cell culture. Interestingly, IgG recognizing S and N was maintained at high positive rates and titers for 6 months. This is particularly important in case of IgG recognizing the RBD of the S protein, since these titers correlated with neutralizing activity and were associated with early virus control, highlighting the relevance of IgG-S as a relevant correlate of protection in humans. It is important that our patient cohort, which showed sustained IgG responses after a transient contraction phase, exclusively comprised symptomatic COVID-19 patients. The time course as well as the duration of humoral immune responses may well be entirely different following asymptomatic infections[6,22].

Considering that severely ill patients had higher IgG-N levels than nonsevere cases at week 4, we speculate that severe COVID-19 patients may experience higher virus replication leading to the expression of more viral antigens, which—maybe in combination with a very strong inflammation—elicit strong humoral immune responses persisting for a prolonged period of time. The hypothesis that antigen levels defined the magnitude of IgG titers may also explain the transient nature of immunoglobulin responses shown for asymptomatic patients.

It is still controversial how antibody titers and the severity of disease may affect each other. One study found that the total antibody levels in severe patients were significantly higher than those of nonsevere patients between the second and fifth week after disease onset, but no differences were observed in IgG or IgM levels alone[23]. Another study observed that the IgG levels of severe patients were significantly higher than in nonsevere patients in the second week after disease onset[3]. The correlation between high antibody levels and severe COVID-19 brought some discussion if antibodies are involved in immunopathology rather than antiviral effects. Contrary to these studies, we found that in the early period following disease onset in nonseverely ill patients and RNA-negative patients (within 3 weeks), the levels of IgM-S/IgM-N/IgG-S/IgG-N were significantly higher than those of severe patients and RNA-positive patients. In addition, there was a clear correlation between IgG-S titers and virus neutralization. This suggests that the antiviral effects of antibodies outweigh potential adverse effects at least during the early phase of COVID-19.

Previous studies have also shown that the plasma of convalescent COVID-19 patients has virus neutralization activity[13] and alleviates symptoms upon administration to severe patients[21]. In agreement with previous studies[12], we found that IgG-RBD-S is positively correlated with neutralizing activity. Some patients showed very high neutralization titers (>1:320) compatible with a status of superior neutralizing capacity. Elite

neutralizers have been described for other viruses like HIV[24]. Obviously, such individuals may enable the identification of broadly and efficiently neutralizing antibody clones, donate superior CP, and promote the design of vaccines raising potent NAb responses[25].

It was discussed that IgG levels of both symptomatic and asymptomatic patients may decrease rapidly during recovery[6], raising concerns about the sustained neutralization activity of patient plasma. Our study demonstrates that the plasma of most symptomatic COVID-19 patients facilitates neutralizing activity during the 6 months observation period, with a considerable proportion of patients exhibiting very high levels of neutralizing activity.

The discussion of rapidly declining humoral immune responses provoked broad media attention, raising doubts and anxiety about the feasibility of vaccine development and immunity after infection. Based on our data, it appears that the humoral immune response to SARS-CoV-2 in symptomatic COVID-19 patients is rather prototypical for viruses in having an early expansion phase followed by an intermediate contraction phase and a sustained memory phase. Analysis that terminated their observation period earlier than in our study, but extrapolated a long-term trend based on the contraction phase without considering or determining the memory/consolidation phase, bear the inherent risk to come to over-pessimistic conclusions concerning the durability of humoral immune responses after SARS-CoV-2 infection. Even primary infections inducing live-long immunity (e.g., measles infection) and very effective vaccine such as the yellow fever and rabies vaccine have a transient contraction phase in the antibody response. Although only the future will show how long protective immunity will last after natural infections or prophylactic vaccination against SARS-CoV-2, our data suggest that SARS-CoV-2-specific antibody responses are quite similar to responses against many other viruses that induce immunity in humans, including the "common-cold" corona viruses that have been shown to mediate protective immunity for many months to years[26,27].

The findings from this work indicate the usefulness of serological antibody tests against S-RBD and nucleoprotein of SARS-CoV-2 for diagnostics. During the early stage of disease, they may be used for the diagnosis of a COVID-19 infection and the levels of antibody responses may be helpful to predict the clinical outcome. In the long term, monitoring antibody levels, especially anti-S-RBD, is beneficial for answering important questions about virus neutralization and immunity against SARS-CoV-2.

This study has some limitations as follows. First, we did not have enough samples at 9–11 weeks because the patients were placed in mandatory isolation for 2 more weeks after discharge from the hospital, followed by another 2 more weeks at home after leaving mandatory isolation. Second, due to the limited availability of the BSL3 laboratory, not all samples could be assessed in virus neutralization tests.

In conclusion, antibodies appear to have antiviral effects in the early stages of SARS-CoV-2 infection, and the most symptomatic patients with COVID-19 remain positive for IgG-S and exhibit sufficient neutralizing activity at 6 months after the onset of illness. These results support the notion that naturally infected patients have the ability to combat reinfection and vaccines may be able to produce sufficient protection. Please note that analyses which terminated their observation earlier than ours and extrapolate the long-term trend based on this contraction phase without considering or determining the consolidation phase bear the inherent risk to come to wrong over-pessimistic conclusions concerning the durability of humoral immune responses.

## Method

**Patients and sample collection**. A total of 585 samples obtained from 349 symptomatic COVID-19 patients from the isolation wards, fever clinics of

Wuhan Union Hospital or National Virus Resource Center of Wuhan Institute of Virology, during the period January 1 to July 15, 2020, were involved in this study. All patients were diagnosed and treated according to the Guidelines of the Diagnosis and Treatment of New Coronavirus Pneumonia (version 7) published by the National Health Committee of the People's Republic of China[28]. All patients met the following conditions: (1) epidemiology history, (2) fever or other respiratory symptoms, (3) typical CT image abnormities of viral pneumonia or decreased lymphocyte count, and (4) positive result of IgG and IgM test or positive result of reverse transcription-polymerase chain reaction (RT-PCR) for SARS-CoV-2 RNA. Severe patients additionally met at least one of the following conditions: (1) low oxygen saturation (≤93%) at resting state or arterial oxygen tension to inspired oxygen fraction (PaO$_2$/FiO$_2$) ≤ 300 mmHg, (2) respiratory failure and requiring mechanical ventilation, and (3) multiple organ failure and admittance to an ICU. We retrospectively collected patient medical records including demographic factors, laboratory results, and other parameters. Individuals coinfected with human influenza A virus, influenza B virus, or other viruses associated with respiratory infections were excluded. Patient blood samples we used to detect antibody levels and neutralizing activity come from the remaining plasma for clinical testing. In order to limit the spread of COVID-19, patients provided verbal informed consent for use of their blood samples instead of signed consent. This procedure was approved by the Ethics Commission of Union Hospital of Huazhong University of Science and Technology in Wuhan. Patients or the public were involved in the design, or conduct, or reporting, or dissemination plans of our research. Blood samples were collected and separated by centrifugation at 3000 × $g$ for 15 min within 4–6 h of collection, followed by 30 min inactivation at 56 °C and storage at −20 °C for further analyses.

**Detection of SARS-CoV-2 RNA and anti-SARS-CoV-2 S/N IgG and IgM**. Throat-swab specimens were obtained from all patients and stored in viral-transport medium for SARS-CoV-2 RNA testing. SARS-CoV-2 RNA was detected by real-time RT-PCR according to the product manual (Daan gene, Zhongshan, China, registration no. 20203400063). Primers targeting the ORF1ab and N genes of SARS-CoV-2 are provided in Supplementary Table 2. IgM and IgG antibodies recognizing the SARS-CoV-2 RBD of the S or the N protein were tested by capture CLIA by MAGLUMI™ 4000 Plus (Snibe, Shenzhen, China) as reported[29]. The cutoff value for IgM-S was 0.7 AU/ml and 1.0 AU/ml for IgM-N, IgG-S, and IgG-N.

**Virus neutralization test assay**. We choose representative samples from groups of patients with different binding IgG titers. Vero E6 cells (1 × 10$^4$ per well) were seeded in 96-well plates one night prior to use. Patients' plasma was incubated at 56 °C for 30 min to inactivate the complement. Twofold serially plasma dilutions in the Eagle's Minimal Essential Medium (NewZongke, Wuhan, China) containing 2% (v/v) fetal bovine serum (Gibco, CA, USA) were prepared. SARS-CoV-2 (Strain BetaCoV/Wuhan/WIV04/2019, National Virus Resource Center number: IVCAS 6.7512) at 100 TCID50 was incubated in absence or presence of diluted plasma for 1 h at 37 °C. Afterward, Vero E6 cell were overlaid with virus suspensions. At 48 h post infection, cytopathic effects (CPE) were visualized and manually judged by microscopic inspection. The neutralizing antibody titer was expressed as the reciprocal value of the highest dilution that prevented CPE formation.

**Statistics and reproducibility**. The mean (standard deviation) was applied for describing continuous variables with a normal distribution, and the median (interquartile range, IQR) was used for continuous variables with a skewed distribution. The difference between groups was examined by Student's $t$ test or Mann–Whitney $U$ test, as appropriate. For categorical variables, $n$ (%) was used for description, and was examined by Chi-square test or Fisher's exact test. Dynamic changes of antibodies tracking from day 1 to day 182 after admission were depicted using the locally weighted regression and smoothing scatterplots (Lowess) model (ggplot2 package in R). Generalized linear mixed models were used for repeated measures statistical analysis. A two-sided $p < 0.05$ was considered statistically significant. The level of statistical significance was depicted as follows: ns, not significance; *$p < 0.05$; **$p < 0.01$; ***$p < 0.001$; ****$p < 0.0001$. All statistical analysis was conducted by R (The R Foundation, http://www.r-project.org, version 4.0.0) and SPSS (version 25, IBM, USA). The precision and reproducibility of the capture CLIA assays were conformed where the same serum samples ($n = 100$) were run independently in two experiments and very similar results were obtained for the independent experiments. Serum samples used for detection of antibody titer were run once tested in our study. Reproducibility of neutralization activity were measured for a subset of serum samples ($n = 50$) and very similar results were seen between the independent experiments. The neutralization activity was tested one time in triplicates for each serum sample in our study.

**Reporting summary**. Further information on research design is available in the Nature Research Reporting Summary linked to this article.

## Data availability
The authors declare that patient data can be provided without names and other identifiers for the purpose of protecting patient privacy. All other data are present in the article and Supplementary files. Source data are provided as a Source data file. Source data are provided with this paper.

## Code availability
All statistical analysis was conducted by R (The R Foundation, http://www.r-project.org, version 4.0.0; ggplot2 package: https://cran.rproject.org/web/packages/ggplot2/index. html) and SPSS (version 25, IBM, USA).

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

## Acknowledgements

This work is supported by the National Science and Technology Major Project for Infectious Diseases of China (2018ZX10302206, 2018ZX10723203, and 2017ZX10304402-002-005); the Applied Basic and Frontier Technology Research Project of Wuhan (2020020601012233); the Fundamental Research Funds for the Central Universities (2020kfyXGYJ016, 2020kfyXGYJ028, and 2020kfyXGYJ046); the Tongji-Rongcheng Center for Biomedicine, Huazhong University of Science and Technology; the Medical Faculty of the University of Duisburg-Essen; and Stiftung Universitätsmedizin Essen, University Hospital Essen, Germany. M.T. was supported by the Kulturstiftung Essen and the Deutsche Forschungsgemeinschaft (DFG) through grants RTG 1949/2, TR1208/1-1, and TR1208/2-1.

## Author contributions

J.W., F.D., U.D., M.T., and X.Z. designed the study. Hua W., Y.F., S.S., B.W., Q.C., Y.W., X.C.Y., W.L., B.Z., W.Z., Huan W., S.M.L., and S.H.L. performed LICA. Y.F., S.S., and F.D. performed neutralization assays. Hua W., Y.H.F. and S.S. prepared pseudovirus or wild-type virus. J.W., B.L., Hua W., X.L.Y., L.K.C., U.D., M.T., and X.Z. analyzed and interpreted the data. Huan W., Y.F., S.S., X.L.Y., Y.W., and J.L. curated hospital serum samples. C.C., Huan W., X.L.Y., B.W., Q.C., and Y.W. assisted in collection of samples from hospitalized patients. J.L., X.C.Y., W.L., B.Z., W.Z., Hua W., S.M.L., and S.H.L. assisted in collection of samples from outpatients. W.L., X.C.Y., F.D., and D.Y. assisted in project administration. J.W., B.L., Hua W., L.K.C., D.L., A.K., M.L., D.Y., U.D., M.T., and X.Z. drafted the manuscript or substantially revised it.

## Competing interests
The authors declare no competing interests.
