## [Peer Review File · Nature Communications]

REVIEWER COMMENTS

Reviewer #1 (Remarks to the Author):

Wu et al. have studied the long-term response to SARS-CoV-2 by analyzing specific IgM and IgG and neutralizing antibodies produced by group of 349 symptomatic patients, followed for up to 6 months. They found that plasma IgM became negative after 12 weeks, but IgG stabilized after a short contraction phase; neutralizing antibodies were present in 70% of the cases. Authors suggest that a sustained humoral immunity exists, which can be an useful indication for vaccine strategies. The paper is interesting and data are presented very clearly.

Comments:

1. in Figure 1A the number of patients studied at each time point could be indicated (like in Fig. 1C).
2. the concept of "elite neutralizers" (lane 224) could be expanded and discussed.

Reviewer #2 (Remarks to the Author):

The authors investigated acute antibody responses to SARS-CoV-2 infection. They found that specific IgM-S/N became undetectable 12 weeks after disease onset in most patients, while IgG-S/N titers showed an intermediate contraction phase, but stabilized at relatively high levels over the six months observation period. High levels of IgM S/N and IgG-S/N at 2-3 weeks after disease onset were associated with virus control and IgG-S titers correlated closely with the capacity to neutralize SARS-CoV-2. Although your studies were well conducted, there was a major concern on the novelty of your work. However, there are some problems that need to be addressed.

1. The main conclusion "While specific IgM-S/N became undetectable 12 weeks after disease onset in most patients, IgG-S/N titers showed an intermediate contraction phase, but stabilized at relatively high levels over the six months observation periods" was based on a total of 585 samples obtained from 349 symptomatic COVID-19 patients. The data will be more convincing if the author analyzed the dynamic changes of IgG and IgM of each patient after onset of symptoms.
2. In Figure 1D, antibody titers of 17 prototypical patients with repetitive sampling were analyzed. The data showed that the majority of patients showed a decline of IgG-S/N 4 weeks after onset of symptoms. This data seems not consistent with the conclusion "IgG-S/N titers stabilized at relatively high levels over the six months observation periods". Therefore, it is important to analyze a large sample size of patients with sequential samples.
3. The author performed virus neutralization tests were conducted using 186 samples from 137 patients. How was this proportion of patients chosen?
4. In discussion, the author mentioned that "we speculate that severe COVID-19 patients experience higher virus replication leading to the expression of more virus antigens". The authors seem to have an opportunity to evaluate the relationship between antibody response and virus replication. Is there any correlation between antibody and clinical course?
5. The paper could benefit from a more in-depth discussion of the potential usefulness of serological assays for SARS-CoV-2.

We proceed with a point-by-point reply:

Reviewer #1 (Remarks to the Author):

Comments:

1. in Figure 1A the number of patients studies at each time point could be indicated (like in Fig. 1C).

R: Thank you very much for this suggestion. Accordingly, we added a data row to depict the number of patients studied at each time point in Fig.1A.

2. the concept of "elite neutralizers" (lane 224) could be expanded and discussed.

R: Thank you very much for this opportunity. We added the following paragraph to the discussion section (at line 293): Some patients showed very high neutralization titers (>1:320) compatible with a status of superior neutralizing capacity. Elite neutralizers have been described for other viruses like HIV²⁴. Obviously, such individuals may enable the identification of broadly and efficiently neutralizing antibody clones, donate superior convalescent plasma, and promote the design of vaccines raising potent NAb responses²⁵.

Reviewer #2 (Remarks to the Author):

1. The main conclusion “While specific IgM-S/N became undetectable 12 weeks after disease onset in most patients, IgG-S/N titers showed an intermediate contraction phase, but stabilized at relatively high levels over the six months observation periods” was based on a total of 585 samples obtained from 349 symptomatic COVID-19 patients. The data will be more convincing if the author analyzed the dynamic changes of IgG and IgM of each patients after onset of symptoms.

R: We thank the reviewer for the good suggestion. We agree that it would be very interesting to study dynamic changes of IgG and IgM in many individual patients after the onset of symptoms. However, it is difficult to get follow-up data from so many patients in China because the patients would be required to visit the local hospital again after discharge, which most patients don't do. In our article and following your advice, we now present the longitudinal antibody results from 35 patients (Fig. 1D), which provides an impression about dynamic changes of IgG and IgM against SARS-CoV-2.

2. In Figure 1D, antibody titers of 17 prototypical patients with repetitive sampling were analyzed. The data showed that the majority of patients showed decline of IgG-S/N 4 weeks after onset of symptoms. This data seems not consistent with the conclusion “IgG-S/N titers stabilized at relatively high levels over the six months observation

periods”. Therefore, it is important to analyzed large sample size of patients with sequential samples.

R: We agree with the reviewer that most patients showed a slow contraction in antibody titers over time. This is actually expected after virus (antigen) clearance. However, with only a few exceptions, all patients had detectable IgG titers even after 26 weeks. To convince the reviewer, we extended the data set from 17 to 35 patients. To optimize the description, we revised the sentence in the result section (at line 171) to: “Except for two unusual patients (patient 13, patient 17), who did not develop measurable IgG-S response, IgM titers generally declined rapidly, while IgG titers were far more stable. Even after 26 weeks, all but three patients (patient13, patient 14, patient 17) presented with detectable IgG antibodies (Fig. 1D, upper panel).”

3. The author performed virus neutralization tests were conducted using 186 samples from 137 patients. How this proportion of patients was chosen?

R: We choose representative samples from groups of patients with different binding IgG titers. We had to do this because not all samples could be tested in the neutralization test. We now describe this in the M&M section at line 449.

4. In discussion, the author mentioned that “we speculate that severe COVID-19 patients experience higher virus replication leading to the expression of more virus antigens”. The authors seem to have an opportunity to evaluate the relationship between antibody response and virus replication. Is there any correlation between antibody and clinical course?

R: We address this question in Figure 2 - in particular in Figure 2C and 2D. We show that patients that experience severe disease show high IgG antibody titers. At 4wpi, the IgG-N titer was significantly higher in severe cases than in non-severe cases (Fig. 2C). This was the basis for our speculation in the discussion.

5. The paper could benefit for a more in depth discussion of the potential usefulness of serological assays for SARS-COV-2.

R: We thank the reviewer for this suggestion. We have added the following paragraph to the discussion section at line 318: The findings from this work indicate the usefulness of serological antibody tests against spike-RBD and nucleoprotein of SARS-CoV-2 for diagnostics. During the early stage of disease, they may be used for the diagnosis of a COVID-19 infection and the levels of antibody responses may be helpful to predict the clinical outcome. In the long term, monitoring antibody levels, especially anti-spike-RBD, is beneficial for answering important questions about virus neutralization and immunity against SARS-CoV-2.

REVIEWERS' COMMENTS

Reviewer #2 (Remarks to the Author):

The author has answered part of my questions, especially question 1, 3 and 4. The main weakness of this study is that the author have only 35 patients with repetitive sampling. This small sample size has weaken the solidation of the conclusion of this study.

We proceed with a point-by-point reply:

Reviewer #2 (Remarks to the Author):

Comments:

The author has answered part of my questions, especially question 1, 3 and 4. The main weakness of this study is that the author have only 35 patients with repetitive sampling. This small sample size has weaken the solidation of the conclusion of this study.

R: We thank the reviewer for the good suggestion. We agree that it will be even more convincing to show more patients with repetitive sampling. Therefore, we added the data set from additional 25 patients to reach a total of 60 patients in Fig. 1d. The overall result that IgM titers declined rapidly in most individuals while IgG titers were far more stable was confirmed and is now based on a solid number of investigated patients.